# Phenolic Compounds in Poorly Represented Mediterranean Plants in Istria: Health Impacts and Food Authentication

**DOI:** 10.3390/molecules25163645

**Published:** 2020-08-10

**Authors:** Ana Miklavčič Višnjevec, Matthew Schwarzkopf

**Affiliations:** 1Natural Sciences and Information Technologies, Glagoljaška 8, Faculty of Mathematics, University of Primorska, SI-6000 Koper, Slovenia; ana.miklavcic@famnit.upr.si; 2InnoRenew CoE, Livade 6, 6310 Izola, Slovenia

**Keywords:** phenolic compound, food authentication technique, health impacts, *Punica granatum* L., *Ziziphus jujuba* Mill., *Arbutus unedo* L., *Celtis australis* L., *Ficus carica* L., *Cynara cardunculus* var. *Scolymus* L.

## Abstract

Phenolic compounds are well-known bioactive compounds in plants that can have a protective role against cancers, cardiovascular diseases and many other diseases. To promote local food development, a comprehensive overview of the phenolic compounds’ composition and their impact on human health from typical Mediterranean plants such as *Punica granatum* L., *Ziziphus jujuba* Mill., *Arbutus unedo* L., *Celtis australis* L., *Ficus carica* L., *Cynara cardunculus* var. *Scolymus* L. is provided. Moreover, the potential use of these data for authenticity determination is discussed. Some of the plants’ phenolic compounds and their impact to human health are very well determined, while for others, the data are scarce. However, in all cases, more data should be available about the content, profile and health impacts due to a high variation of phenolic compounds depending on genetic and environmental factors. Quantifying variation in phenolic compounds in plants relative to genetic and environmental factors could be a useful tool in food authentication control. More comprehensive studies should be conducted to better understand the importance of phenolic compounds on human health and their variation in certain plants.

## 1. Introduction

Only 20% of the food consumed in the European Union is sourced locally despite the positive benefits for local economies. Moreover, regional foods, as a result of regional traditions and history, have the potential to promote and attract tourism [1,2]. In order to support the local food development of the EU and beyond, policies can be developed on the federal, state, and local government levels to help support this effort. For example, the United States is trying to support local food by addressing issues in their policies such as scaling up output of small farms, education of producers, improving infrastructure and the ability to trace where certain products come from [3]. In a study performed by Willis et al. [4] the results suggested that local food banks, transportation, storage and processing centers could provide resources to strengthen food distribution networks and therefore local farms could successfully compete. Despite many attempts to promote local food usage, the main issues include difficulties of smaller farms to compete on larger markets due to a lack of price competitiveness, limited available product volume and logistic considerations [5,6].

One method used to promote local resources is through designation or origin of various products. By designating the region a product comes from, certain specialties or traits consistent with that region can be highly valued. The designation of origin is one method widely used in Mediterranean food products, including those in Istria. Many of the food products produced in this region are prized for their good taste and positive human health attributes. Some of these health benefits can be linked to the presence of phenolic compounds which are defined as compounds containing hydroxylated aromatic rings. It is well known that phenolic compounds can have beneficial effects against cancer, cardiovascular disease and many other diseases [7,8,9,10,11,12,13,14,15]. The positive effects of phenolic compounds on various chronic degenerative diseases, cardiovascular diseases and cancers [7,8,9,11,15,16] could be due to their antioxidant efficacy, lowering the amount of free radicals within the body [16]. Free radicals can cause damage to cells and their constituents that can lead to the onset and development of chronic degenerative diseases. In addition, antioxidants inhibit the action of enzymes that catalyze the detoxification pathway in the body upon exposure to mutagens [17]. Based on their bioactive effects, phenolic compounds can also contribute to the quality of food.

In addition to their antioxidant effects, Crozier et al. [16] described in detail the exertion of different specific modulatory effects of phenolic compounds in cells important for cellular growth, proliferation and apoptosis. Therefore, their mechanism of action does not involve only the antioxidant activity, but also the selective actions on various components of the intracellular signaling cascades that are vital for cellular functions [16].

With these high-quality attributes, it is important for local farmers to label their products accordingly and ensure that they have legitimate claims to their designation of origin. Phenolic compounds are not only important for their positive impacts on heath. Due to their high variation according to the environmental and genetic factors, they can be used as a possible research authenticity approach, as demonstrated effectively in different studies [18,19,20,21,22,23].

Due to the adulteration of food products in large markets, food traceability and effective food authenticity control systems are becoming important in relation to both consumer protection authorities and producers and dealers [24,25]. In this regard, food authentication, defined as “the process that verifies a food is in compliance with its label description” [26], is a rapidly growing field. The most successful techniques for determining the authenticity of foodstuffs, described in detail by Primrose et al. [27], involve: (1.) use of stable isotope analysis; (2.) methods based on DNA analysis; (3.) proteomic methods and (4.) metabolomics methods. In addition, there are many other possible research authenticity approaches including high-performance liquid chromatographic (HPLC) in combination with mass spectrometry (MS) of phenolic compounds in different foodstuffs [18,19,20,21,22,23]. However, due to different products on the market and an increasing number of new technologies being used, the analytical determination of authenticity including standardized and harmonized operating procedures as well as quality assurance measures is a challenging task [25].

In this review, the phenolic compounds composition of typical Mediterranean plants is addressed. These plants are well adapted to growing conditions in Slovenian Istria, but they are currently poorly represented in the landscape and habitats in this area, due to the increasingly global market and difficulties that smaller farmers have to compete with in a global market. This review focuses on plants that are not only important for the revitalization of species and preservation of natural heritage but also possess bioactive compounds such as phenolic compounds [28,29,30,31,32,33,34,35] that can have a beneficial effect against various diseases [7,8,9,10,11,12,13,14,15] and are important for human nutrition, due to high mineral, vitamin and dietary fiber [7,15,29,30,36,37,38,39,40,41,42,43,44] content. In order to promote local food development, a comprehensive overview of the phenolic compounds’ composition and their impact on human health is provided. In addition, the potential use of this data for authenticity determination is discussed. This review emphasizes the economic importance of these plants and provides directions for future studies.

## 2. Phenolic Compounds from Different Plant Sources

Bioactive compounds are secondary metabolites in plants known for their pharmacological or toxicological effects in humans and animals [45]. According to Croteau et al. [45], bioactive compounds in plants are divided into three main categories: (1.) terpenes and terpenoids; (2.) alkaloids and (3.) phenolic compounds. Flavonoids or other phenolic compounds, including tannins, form one of the most represented groups of bioactive compounds in fresh fruits and vegetables [16]. By definition, the term phenolic compound involves a wide range of various solvent-soluble compounds, having at least one aromatic ring with one or more hydroxyl groups attached. Based on their structural skeletons, these compounds are broadly divided into phenolic acids, acetophenones, phenylacetic acids, hydroxycinnamic acids, coumarins, naphthoquinones, xanthones, stilbenes and flavonoids [16]. Flavonoids are the most abounded compounds found in plants for human consumption. Among the subclasses of dietary flavonoids, the flavonols, flavan-3-ols, flavones, flavanones, isoflavones and anthocyanidins are the most represented in human nutrition, while other groups are minor components of the human diet [16]. The basic flavonoid ring may have numerous substituents. In nature, flavonoids are very often in the form of glycosides, consisting of up to three monosaccharides. While the binding of sugars and hydroxyl groups increases the solubility of flavonoids, other substituents characteristics for flavonoids make flavonoids lipophilic [16].

Hydroxycinnamic acid is also frequently found in nature and largely distributed in different plants [46]. They are formed from phenlyalalanine and tyrosine. Their basic skeleton is C_6_C_3_. The most common and best known hydroxycinnamic acids are cinnamic acid, *o*-coumaric acid, *p*-coumaric acid, caffeic acid, ferulic acid and synaptic acid [46,47,48]. They are potential antioxidants that can have effects on a large number of diseases such as cancer and atherosclerosis [48].

In addition to flavonoids and hydroxycinnamic acids, elagitannins and gallotanins were present in selected plants. Freudenberg divided plant tannins into two groups: (1) condensed tannins and (2) hydroliyzable tannins. Gallotannins and elagitannins are subclasses of hydrolysable tannins and they are derivatives of 1,2,3,4,6-penta-*O*-galloyl-B-D-glucopyranose [49].

Phenolic compounds are produced in a plant’s cells and may protect the plant against pests and other stress factors. In addition, they contribute to the color and taste of fruits and vegetables, and, at higher concentrations, they protect plants from attack by viruses, bacteria, fungi and herbivorous animals. In the case of plant or pest infestation, the genetically determined formation of phenolic compounds in a plant can greatly increase [45]. It is important to emphasize that the content of phenolic compounds in a particular plant can vary significantly, and the concentration can be highly dependent on both genetic and environmental factors. Therefore, the content of phenolic compounds can be influenced by cultivar selection and conditions present in the growing area. It is well known that concentrations, even within a single fruit, can vary significantly and that, in addition to the other environmental factors, the phenolic compound content in fruits and vegetables can be affected by growing technology [50]. Insights that can be obtained from epidemiological studies about the beneficial effects of bioactive compounds on human health are not only limited by the complex variation of these compounds, depending on many different factors, but also by the lack of comprehensive and reliable data for the phytochemical content of raw food. The analysis of secondary metabolites in plants, estimated to be approximately 100,000 to 200,000 in number, is a challenging task because of their chemical diversity, which is usually low in abundance and variability even within the same species [51].

## 3. Phenolic Compounds and Their Impact on Human Health

### 3.1. Pomegranate

The pomegranate (*Punica granatum* L.) belongs to the Lythraceae family [52], and it is believed to originate from an area between Iran and the Himalayas in northern India. Pomegranates have been grown in the Mediterranean since ancient times. The fruit is consumed fresh or used for the production of juices, syrups, jams and wines [53]. Pomegranate is a rich source of bioactive compounds and minerals found in its peel, seeds and arils [7,36,54]. It contains known amounts of phenolic compounds, including catechins, anthocyanins and other complex flavonoids, and hydrolysable tannins (punicalagin, punicalin, pedunculagin, gallic acid and elagic acid) (Appendix A). These compounds are found in pomegranate peel and juice as well and amount to approximately 90% of the antioxidant activity determined in the whole fruit [55,56,57]. The predominant phenolic compounds are punicalagin, gallic acid and elagic acid [55,58]. Due to the polarity of methanol, methanol extracts of pomegranate peel showed the highest antioxidant potential compared to other solvents. However, aqueous extracts demonstrated better anti-tumor activity compared to methanol extracts [57,59,60,61].

Numerous studies have shown the beneficial effects of pomegranate against various diseases such as cardiovascular diseases, cancers, skin diseases, diabetes and oral cavities as well as possible positive effects against obesity, male infertility, arthritis, Alzheimer’s disease and diarrhea [7,8]. The extracts of pomegranate peel showed the ability to inhibit inflammation and allergies [62]. The anti-inflammatory compounds of the pomegranate peel, such as punicalin, granatin B, punicalagin, and strictinin A significantly reduced the production of nitric oxide and prostaglandin-E2 by inhibiting the expression of proinflammatory proteins [63,64]. Natural antioxidants such as flavonoids can inhibit the formation of free radicals [65,66] that can contribute to the development of cancer diseases. In addition, phenolic compounds in pomegranate can affect the growth of cancer cells by induction of apoptosis. Therefore, the cytoprotective and inhibitory effects of pomegranate extracts have the potential to prevent certain kinds of cancer [55].

Due to the wide use of pomegranate extracts as a food supplement, it is very important to determine the toxic effect of chronic or sub-chronic consumption of these extracts. Pomegranate extracts are a rich source of phytochemicals that can cause toxic effects at higher concentrations after long term consumption [56,67]. However, it has been shown that the moderate consumption of pomegranate juice or extracts in a specific concentration range is safe [55]. Even though numerous studies exist on bioactive compounds in pomegranate, more research should be carried out to fully understand their mode of action and their preventive and therapeutic potential.

Overall, the demonstrated beneficial effects of phenolic compounds such as granatin B punicalagin, punicalin and strictnin A or many others phenolic compounds found in pomegranate can be very important for promoting local food in Istria.

### 3.2. Jujube

The jujube (*Ziziphus jujuba* Mill.) belongs to the Rhamnoceace family [52] and grows in a warm climate. The plant originates from China. Cultivation of the jujube was first spread to neighboring countries of China and then to Europe, most probably via the Silk Road [68]. The fruit can be enjoyed fresh or processed in jams, jellies and pastries. Jujube fruit contains high levels of sugar, and it is a rich source of vitamins C, A and B-complex as well as minerals such as phosphorus and calcium [37]. Other important compounds in the jujube fruit are unsaturated fatty acids, sterols, alkaloids, tannins, flavonoids and saponins [28]. The health benefits of jujube fruits include anti-inflammatory, immunostimulating, anticancer, anti-obesity, hepatoprotective and gastrointestinal protective activities [9]. Recent studies have shown that polysaccharides in the jujube fruit are one of the major biologically active compounds with immunomodulatory, antioxidant, antitumor, hepatoprotective and hypoglycemic properties [10]. In addition, the concentrations of the total phenolic compounds determined by the Folin–Ciocalte method (275.6–541.8 mg GAE/100 g) are higher in jujube [30] compared to the other fruits that are well known for high phenolic compounds content such as the cherry (114.6 mg GAE/100 g), apple (74.0 mg GAE/100 g) or red grape (80.3 mg GAE/100 g) [69]. Our previous work confirmed that the jujube from Slovenian Istria is a good source of phenolic compounds [30] with known pharmacological properties [9,30]. The concentration of total phenolic compounds in the peel could be from five to six times higher than in the jujube mesocarp [70]. However, it should be noted that the content of phenolic compounds varies according to different jujube varieties such as Tuanzao, Junzao, Fegmiguan, Jiaxinmuzao, Taigumizao, Lingbaozao, Zaowangzao, Puaisanhao, Jinchangyhao and Qingjianmuzao. In addition to genetic factors, it was also shown that the altitude and amount of annual precipitation significantly affect the phenolic compounds content. Jujube fruits grown in severe drought and at high altitudes had higher concentrations of phenolic compounds than jujube fruits grown in lower areas and in mild environmental conditions [71]. Phenolic compounds such as flavonoids could vary according to the maturity stage of the jujube fruit as well [72,73,74]. Different kinds of phenolic compounds that are presented in Appendix A, such as of flavonols and flavan-3-ols were found in jujube. More specifically, the following flavonoids were determined: kaempferol-glucosyl-3-ramnoside, quercetin-3-rutinoside, procyanidin B2, catechin, quercetin-3-galactoside, glucosyl-ramnoside, epicatechin and quercetin-3-robinobioside [72,73,75]. Wang et al. [76] reported the following phenolic acids: ferulic acid, caffeic acid, coumaric acid, cinnamic acid and chlorogenic acid. They determined that the concentrations of phenolic compounds in jujube fruit range from 751.39 ug/g w.w. in the peel to 143.59 μg/g w.w. in the mesocarp. Most of the determined phenolic compounds were in glycosid form (44.7%); *p*-hydroxybenzoic acid prevailed both in the jujube mesocarp and seeds while the peel consisted predominantly of chlorogenic acid, *p*-coumaric acid and cinnamic acid [76]. Phytochemical data in combination with biological activities indicate the potential value of jujube in medicine and human nutrition. However, unidentified phenolic compounds or other compounds, including saponins and non-polar compounds, should be further investigated [77].

The prevalence of flavonoids and certain phenolic acids among phenolic compounds in jujube fruit are valuable compound together with other bioactive compounds found in jujube fruits for promote broader local use.

### 3.3. Strawberry Tree

The strawberry tree (*Arbutus unedo* L.) belongs to the family Ericaceae [52], and it originates in the Mediterranean area. Fresh fruits are used for the production of alcoholic drinks and jams. Strawberry tree fruits are a good source of vitamins and minerals, in particular calcium and potassium. They also contain a lot of sugars [38,39,40]. A number of the strawberry tree’s beneficial effects on human health such as beneficial effects on hypertension, cardiovascular diseases and diabetes could be the consequence of the content of antioxidants such as phenolic compounds [11]. Various phenolic compounds belonging to different groups that are presented in Appendix A have been determined in strawberry tree fruits such as galloyl derivatives, phenolic acids, flavan-3-ols, flavonols and anthocyanins [40,78]. Ayaz et al. [40] determined the composition of strawberry tree fruits’ phenolic acids from Samusun in Turkey. Gallic acid predominated, followed by protocatechuic acid, gentisic acid, *p*-hydrobenzoic acid and *m*-anisic acid. In the strawberry tree fruits from the Caceres region (Spain), the following phenolic compounds were determined by HPLC:, procyanidin (474.1 mg/100 g d.w.), catechin (313.4 mg/100 d.w.), hydroxybenzoic acid (112.2 mg/100 g d.w.), ellagic acid (6.9 mg/100 g d.w.), anthocyanins (5.8 mg/100 g d.w.), flavonols (3.6 mg/100 g d.w.) and hydroxycinnamic acid (1.0 mg/100 g d.w.). The main phenolic compounds determined in wild strawberry tree fruits from Portugal by HPLC-ESI-qTOF/MS were flavan-3-ols and galloyl derivatives (60.93 mg/100 g), followed by anthocyanins (13.77 mg/100 g) and flavonols (10.89 mg/100) [79]. Guimarães et al. [79] further identified the following compounds: quercetin-3-*O*-rutinoside, quercetin galloylhexoside derivatives, kaempferol hexoside, B-type procyanidin tetramer, myricetin rhamnoside, quercetin pentoside, quercetin rhamnoside, and quercetin-3-*O*-glucoside.

Within the group of flavan-3-ols and galloyl derivates, strictinin elagitannin, galloylquinic acid, B-type proanthocyanidin trimers, galloylshiquimic acid, B-type procyanidine dimer, galloylhexoside acid, digalloylquinic acid, digalloylquinic shikimic acid and (+)-catechin were identified. The following anthocyanins were identified and quantified: cyanidine-3-*O*-pentoside, delphinidin-3-*O*-glucoside and cyanidine-3-*O*-glucoside.

Mendes et al. [80] identified the following phenolic compounds in the strawberry tree fruits from Portugal: strictinin ellagitannin, digalloylquinic acid, gallic acid glucoside, galloylquinic acid, quercetin glucoside, quinic acid derivative, trigalloylshikimic acid, proanthocyanidin dimer, galloylshikimic acid, digalloylshikimic acid, catechin monomer, proanthocyanidine trimer, ellagitannin derivative, galloyl derivative, myricetin rhamnoside, gallotannin and ellagic acid rhamnoside. Fortalezes et al. [81] identified similar phenolic compunds. Some of the anthocyanins found in strawberry tree fruits from Salamanca (West Spain) [82] were different from those found in strawberry tree fruits from Portugal [79]. Spanish fruits contained cyanidin-glucoside, cyanidin-galactoside, cyanidin-arabinoside and delphinidin-3-galactoside In the study from Spain, cyanidin-galactoside was present at the highest concentrations (2.8 mg/100 g edible portion), whereas in the Portuguese study, cyanidin-3-*O*-glucoside (11.40 ug/100 g d.w.) was prevalent. Besides the different environmental and genetic factors between the two studies, the differences could be due to a different way of quantification and extraction. In the Portuguese study [79], an external standard was used for every anthocyanin; while in the Spanish study [82], only one calibration curve obtained with the cyanidin-3-glucoside was used. There were also some differences in the levels of flavonols between the two studies.

In the Spanish study, Pallauf et al. [82] identified myricetin-3-*O*-xyloside and quercetin-3-*O*-xyloside that were not determined in the Portuguese study [79]. The phenolic compounds found in the strawberry tree fruits from Pisa in Italy included anthocyanins (cyanidin-3-*O*-glucoside, cyanidin-3-*O*-arabinoside and delphinidin-3-*O*-galactoside), 5-*O*-galloylshikimic acid, 3-*O*-galloylquinic acid, gallic acid-4-*O*-β-D-glucopyranoside, 5-*O*-galoylquinic acid, β-D-glucogalline, 4-arbutin and 3-*O*-galloylshikimic acid [12].

Different results of quantified and identified phenolic compounds in the strawberry tree fruits from different areas demonstrate the importance of studying the bioactive compounds in potentially economically important plants in different areas and the possibility of using the phenolic compound fingerprint for determination of geographical origin for food from these plants.

Even though studies on *A. unedo* health effects are lucking, almost all part of *A. unedo* are used in traditional medicine [14]. These demonstrate the importance of further studies and the promotion of this plant for broader local use.

### 3.4. Mediterranean Hackberry

Mediterranean hackberry (*Celtis australis* L.) belongs to the Cannabaceae family [83] and grows in the Mediterranean area and in parts of South-East Asia [84]. The fruits are rarely enjoyed fresh. Usually, they are used for the production of liqueurs [85]. Mediterranean hackberry fruits contain a lot of sugars and they are rich in fiber, vitamins, minerals and phenolic compounds [41,42]. Different phenolic compounds identified in Mediterranean hackberry are shown in Appendix A. The leaves contain rare phenolic compounds such as cytisoside, acacetin 7-*O*-glucoside and isovitexin [31,32,33] (Figure 1). Extracts of the fruits and leaves have a positive effect against diarrhea, menstrual bleeding, amenorrhea, colic, dysentery and peptic ulcers [13]. In our previous study [42], the preliminary results of nutritional values, mineral composition and content of phenolic compounds in Mediterranean hackberry fruits (mesocarp and seeds) from Istria were presented for the first time. Seasonal differences in the content of minerals and total and individual phenolic compounds were observed. Ethanol extracts and aqueous leaf extracts showed antimicrobial efficacy. A preliminary study indicated the need for further identification and determination of bioactive compounds in the leaves and fruits of this plant [42]. In addition, further studies are needed to determine the health impact of this plant. However, the wide use of Mediterranean hackberry in traditional medicine is a good indicator for promoting the growing and marketing of the plant for certain areas.

### 3.5. Fig

The fig (*Ficus carica* L.) is a widespread species of the Moraceae family. It grows in warm and dry climatic conditions, mostly in the Mediterranean area [43]. In the past, the fig from the part of Slovenian and Croatian Istria with a Mediterranean climate had greater economic significance than today. Fruits are consumed fresh, dried or processed in jams, marmalades and compotes [86]. Figs are a good source of minerals such as iron, calcium and potassium, vitamins, dietary fiber and contain a large number of different amino acids [87,88]. They also contain high levels of phenolic compounds known to have positive effects on human health [16]. The phenolic compound profile found in figs are shown in Appendix A. Solomon et al. [44] determined anthocyanins in the skin and fruit pulp from commercially available figs of different varieties and colors. Dueñas et al. [89] characterized, quantitatively and qualitatively, the composition of skin and pulp pigments for various commercial varieties of Iberian figs. Fifteen species of anthocyanin pigments were found, most of them containing cyanidin as aglycone and some pelargonidine derivatives. Rutinose and glucose were present as substituted sugars. Acylation with malonic acid was also present. Small amounts of peonidin-3-rutinose were identified in the fig pulp. In addition, the authors determined the following phenolic compounds: cyanidin 3-rutinose dimer, 5-carboxypyranocyanidin-3-rutinoside, and five condensed pigments containing C-C linked catechin, anthocyanins and epicatechin residues [89]. From the literature, it is known that antioxidant activity correlates with certain determined levels of phenolic compounds in figs [44]. In comparison to red wine (200–800 mg/200 mL) or black tea (150–210 mg/200 mL), which are known for their high content of phenolic compounds, figs contain higher concentrations of total phenolic compounds (1090–1110 mg/100 g w.w.) [87,88]. Figs are not just a high-nutritional food but are also used in traditional medicine, homeopathic medicinal products and nutritional supplements of plant origin that are widely used worldwide [88]. Since data on the beneficial effects of the fig on human health are based primarily on folklore data and anecdotes [88], further studies in this field are very important and promising. Overall, figs are a good source of minerals, vitamins, dietary fiber, amino acids and phenolic compounds such as anthocyanins that have a potential impact on human health and therefore are important to enhance the growing of this plant in certain Mediterranean areas.

### 3.6. Globe Artichoke

The globe artichoke (*Cynara cardunculus* var. *Scolymus* L.) belongs to the Asteraceae family, and it is a Herbaceous perennial plant with widespread cultivation in the Mediterranean area [90]. Globe artichoke is a rich source of minerals and fiber [15]. The leaves are known for their beneficial and therapeutic effects on human health, including improving blood circulation, mobilizing energy stores, inhibiting cholesterol biosynthesis and oxidizing LDL cholesterol. It has important antioxidant, anti-cancer and hepatoprotective effects [91,92]. Many studies have linked the positive effects of artichoke on human health to the content of phenolic compounds, such as cynarine, mono- and di-caffeoylquinic acid, luteolin and its 7-*O*-glucoside [15,93]. Pandino et al. [93] quantified the phenolic compounds in different parts of the artichoke head by HPLC-ESI-QqQ, and the authors determined the presence of: luteolin, 5-caffeoylquinic acid, apigenin-7-*O*-glucoside, 3-5-dicaffeoylquinic acid, 1-caffeoylquinic acid, 3-caffeoylquinic acid, luteolin-7-*O*-rutinoside, luteolin-7-*O*-glucoside, luteolin-7-*O*-glucuronide, apigenin-7-*O*-rutinoside, hesperetin, 1,5-dicaffeoylquinic acid, apigenin-7-*O*-glucuronide, apigenin and malonyl-glucoside (Appendix A).

The profile of phenolic compounds in the globe artichoke was significantly different between different varieties and in various parts of the head, suggesting that individual compounds are accumulated preferably in certain parts of the head and in specific varieties. The concentrations of phenolic compounds in the outer bracts were much lower (443 mg/kg w.w.) compared to the receptacle (473 mg/kg w.w.) [93]. Brat et al. [94] showed that the edible part of the artichoke head had the highest concentration of phenolic compounds among twenty-nine different vegetables from France included in the study. The studies described above prove the importance of researching globe artichokes from Slovenian Istria that could have a specific composition of phenolic compounds. However, further studies are needed to better evaluate the health effects. Overall, in addition to the fact that global artichokes are rich sources of minerals and fiber, they contain significant amounts of various phenolic compounds that are important for the promotion of broader local use.

## 4. Food Authentication Technique

Danezis et al. [26] classified different analytical techniques for authenticity assessment based on authenticity indicators: (1.) Molecular techniques, genomics–proteomics; (2.) Chromatographic techniques; (3.) Isotopic techniques; (4.) Vibrational and fluorescence spectroscopy; (5.) Elemental techniques; (6.) Nuclear magnetic resonance (NMR); (7.) Sensory analysis; (8.) Non-chromatographic mass spectrometry and (9.) Immunological techniques. Molecular techniques such as nucleonic- and proteomic-based methods for food authentication are generally used for species and botanical origin detection and identification. All of the other techniques are also focused on geographical origin and adulteration [26]. Mass spectrometry (MS) techniques are rapidly replacing other methods in food authentication because they are selective, sensitive and can provide multi-analyte results [26,95]. Food authentication determination based on chromatographic techniques coupled to MS involving small molecules such as phenolic compounds is further discussed in the next paragraph.

### Phenolic Compounds Characterization for Determination of Authenticity

Chromatographic methods coupled with MS can differentiate and authenticate foods based on minimal analytical differences between patterns or identification of unique marker compounds Liquid chromatography can target chemical compounds by polarity, electrical charge and molecular size, and it is generally used in addition to detecting phenolic compounds and other groups of compounds such as triglycerides, proteins, vitamins, pigments and chiral compounds [26]. Phenolic compound analyses determined by HPLC-MS may be broadly divided into targeted and untargeted applications. In the case of targeted analyses, a finite number of known target molecules can be determined, mostly quantitative; while in the case of non-target or screening analyses, the main goal is to identify all possible phenolic compounds, in most cases only qualitative. For the latter, the most appropriate is qTOF (time-of-flight) mass spectrometers that offer very high mass ranges without having to compromise sensitivity. Hybrid TOF instruments, such as quadrupole-TOF mass spectrometers (qTOF), allow collimation of the ion beam as well as the option of mass selection in the first quadrupole and fragmentation in a quadrupole [96]. Such research equipment makes it possible to compare samples based on their similarity and differences in a semi-automated and untargeted manner. It also provides accurate analysis for a wide range of metabolites with different polarities compared to the standard liquid chromatography method [97]. TOF mass spectrometers have better performance, although not in regards to sensitivity, to triple quadrupole mass spectrometers (QqQ) [96,98]. In addition, other mass spectrometry techniques such as “fast atom bombardment mass spectrometry” (FAB-MS), “matrix-assisted laser desorption/ionization mass spectrometry” (MALDI-MS) and electron impact mass spectrometry have also been used for structural identification and confirmation of phenolic compounds [99]. Specific analytical equipment such as liquid chromatography coupled with MS is essential for the characterization of plant extracts. Such equipment allows for the identification of many bioactive compounds at the same time at very low concentrations [96,100] and could be used for the determination of the authenticity of food.

Because fruits’ phenolic compounds composition shows different profiles and concentrations according to species, cultivar and ripening stage, the characterization of phenolic compounds could provide a method for authenticity control of fruit-based products [101]. Obon et al. [102] determined 58 compounds after analyzing nine different red fruit and vegetable juices by HPLC UV-Vis and fluorescence detectors. Every red fruit or vegetable juice involved in the study showed a characteristic anthocyanin or betacyanin profile. These analyses were useful for verifying the authenticity of fruit and vegetable juices [102]. Wen and Wrolstad [103] determined the composition of authentic pineapple juice and concluded that some of the detected compounds may serve as marker compounds for pineapple juice. Dragovic-Uzelac et al. [104] proved that HPLC profile of phenolic compounds could be used as a “fingerprint” for controlling the authenticity of apricot purees, nectars and jams. Silva et al. [21] determined the phenolic compound profile to evaluate quince jelly’s authenticity. By not detecting arbutin, the authors could conclude that none of the samples were adulterated by the addition of apple or pear. By HPLC phenolic compounds fingerprint and sophisticated statistical analysis, Cavazza et al. [18] could differentiate honeys by botanical origin with a high degree of agreement. The phenolic composition was also studied in different varieties of Bauhinia forficata in order to use the data for authenticity control by HPLC-DAD-MS^n^ [19]. The authors concluded that the authentication of Bauhinia samples by their flavonoid profile was successful. Similarly, Malacarne et al. [20] checked the authenticity of commercial tannins and concluded that verifying the botanical authenticity of commercial tannins was only possible when simple phenolic compounds were included in addition to the profiling of carbohydrates and polyalcohols. Marova et al. [23] identified 49 phenolic compounds in different samples of lager beers and three alcohol-free beers by LC-PDA-ESI-MS and found a representative set of flavonoids with the potential to be used in order to determine the authenticity of Czech beer. Further studies are needed in combination with other methods for authenticity determination and ambiguous evaluation of authenticity for certain fruit and vegetable products. As was previously reported by Danezis et al. [26], the main challenge is to evaluate the large volumes of data generated by these high-performance analytical techniques.

Similar to the studies described above, the phenolic compounds described in poorly represented Mediterranean plants could be used for authenticity determination. However, it must be noted that it is not possible to make assumptions about which species can be used to authenticate different food based only on data about the phenolic compounds provided in the literature. Specifically, the different extraction and sampling methods make it very difficult to compare phenolic compounds data reported in the literature. However, rare phenolic compounds present only in certain plants such as acacetin 7-*O*-glucoside, isovitexin and cytisoside, which can be marker compounds for the presence of Mediterranean hackberry in processed food or supplements, or punicalin, pedunculagin and punicalagin can confirm the presence of pomegranate in processed food such as pomegranate juice. In addition, cynarine could be a good indicator of the presence of artichoke. Furthermore, this review emphasizes the need for further studies analyzing phenolic compounds by sophisticated analytical methods such as HPLC-MS/MS combined with statistical methods for determination of phenolic compound species unique profiles.

## 5. Conclusions

Nowadays, access to sufficient, safe and nutritious food is very important. In this regard, it is crucial to enhance the revitalization of plant characteristics in specific areas that represent an important role in human nutrition. This review addressed poorly represented Mediterranean plants in Istria that are rich in phenolic compound content and have economic potential for the area. Although some of the discussed plants are very well described in the literature regarding phenolic compounds and their impact on human health, more data should be made available due to high variation in these compounds, which are highly dependent on both genetic and environmental factors. On the other hand, high variation of the content and profile of phenolic compounds that depend on certain factors could be a useful tool in authentication control regarding the botanical and geographical origin of fruit and vegetable products discussed in this study. By using this authentication tool, the promotion of local food products can be supported through the responsible marketing of their positive human health impacts like those from phenolic compounds, with their designation of origin. Furthermore, the analysis of these compounds from plants in certain areas is needed, due to a high variation of these compounds presented in this overview, in order to enhance, promote and valorize local food and its products.

## Figures and Tables

**Figure 1 molecules-25-03645-f001:**
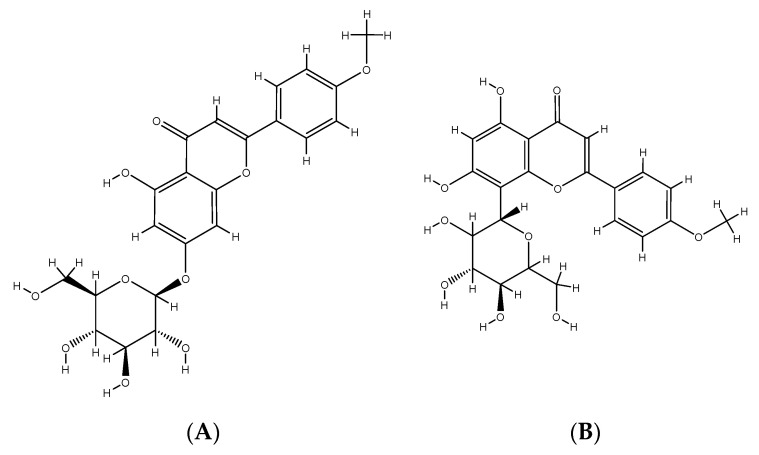
Rare phenolic compounds such as acacetin 7-*O*-glucoside (**A**), and cytisoside (**B**) and isovitexin (**C**) identified in Mediterranean hackberry (*Celtis australis* L.) leaves.

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
