# Peer review of "Phenolic Compounds in Poorly Represented Mediterranean Plants in Istria: Health Impacts and Food Authentication"

_molecules, 2020, doi:10.3390/molecules25163645_

Round 1
Reviewer 1 Report
There is a large set of information in this review that has the potential to be widely useful and offer an important direction for how profiling phenolics could benefit the food industry and health.
My major concern is the lack of integration of phenolics in general, their health benefits, use for authentication and why this integration is important. As written, the review is really just several chunks of information that are not connected to larger ideas.
I suggest a major restructure of the paper - much of the information is already there, but it does not tell a compelling narrative for a reader. Here is my recommended restructure:
Intro:
1. Need for increased local consumption. Set up problem and why important. Current solutions, but what they lack.
2. Need for authentication. Set up problem and why important. Current solutions, but what they lack.
3. Identify phenolic profiles as way to contribute to both. What are phenolics (lines 66-100)
4. Demonstrate they Have health benefits (lines 101-306) - use knowledge to promote broader local use.
5. Highlight which species unique profiles (from lines 101-306 and associated tables) could be used to authenticate (inclusion of particulate plants in common foods and region they came from - (e.g. ,Spanish vs Portugal distinction)).
4 summarize with how knowing phenolics promote health benefits of foods originated from specific regions.
In addition to this restructuring, I offer recommendations for better clarity of ideas as comments in yellow in the original submitted PDF.

Reviewer 2 Report
Phenolic compounds are well-known bioactive compounds in plants that can have a beneficial effect against cancer, cardiovascular disease and many other diseases. In order to promote local food development, the present review outlined 6 poorly represented Mediterranean plants (Pomegranate, Jujube, Mediterranean hackberry, Hackberry, and Globe artichoke) in Istria that are rich in phenolic compounds content and have economic potential for the area. For each plant, the phenolic compounds’ composition with the identification methods and their impact on human health were reviewed and summarized. Moreover, the potential use of these data for authenticity determination is further discussed. There are some concerns as listed in the following:
*L23: It is better to include these keywords (Punica granatum L.; Ziziphus jujuba Mill.; Arbutus unedo L.; Celtis australis L.; Ficus carica L.; Cynara cardunculus var. scolymus L.) in the Abstract.
L57: diseases [13--21]
L59: dietary fiber [24, 25, 13, 21, 31-39] -> Ref. order
L75: and flavonids-> and flavonoids
*L103: (Punica granatum L.)-> (Punica granatum L.) -> Plant name uses italic letter. (Check all)
*L107: aryls-> arils
*L121: prostaglandin-2?
*L134: In Table 1-6, it is better to use a line to separate the results from different References.
L152: phenolic compounds varies according to the variety [64].? -> specify which variety
*L379: Check all to keep one format for the writing of Journal, e.g. R6: Trends Analyt Chem. vs R8: Trends Food Sci Technol vs. R14: Journal of Agricultural and Food Chem vs. R16: Compr. Rev. Food Sci. Food Saf. vs. R18: International Int. j. res. dev. pharm. life sci. vs. R44: Biochemistry and molecular biology of plants vs. R53: J. Agric. Food Chem vs. R97: Research in Mass Spectrometry
Reviewer 3 Report
This manuscript majorly reviews the phenolic compounds, identified from Mediterranean plants and their health impacts. Several plants were involved, and the composition of phenolic compounds was summarized in table 1 to 6. In addition, the way to identify these compounds was also mentioned brief afterwards. As far as I am concerned, this manuscript can be assigned as major revision.
Here are my questions and concerns.
- Page 2, line 73-79. Here is a brief summary of the category of phenolic compounds, however, most of the sentences focused on the structure of flavonoids. However, in Table 1-6, besides flavonoids, a big quantity of compounds was not only flavonoid, but also phenolic acid derivative and coumarins. Therefore, it is highly recommended to add a detail category of phenolic acids, paired with some structures. If possible, group the compound in table 1-6 based on the category mentioned. Otherwise, the function of table 1-6 is merely listing the names only, which is not helpful for understanding.
- Page 3, line 101. The title does not fit. You might want to say, Phenolic compounds from different plants sources.
- Also, the title of the manuscript mentioned the impact on human health, however, there were not many related content in the review, or just mentioned it has health impact directly. It is highly recommended to add some details after each section of the plant sources. For example, the phenolic compounds identified from pomegranate displayed biological activity, such as cytotoxicity, antioxidant, etc… The examples shown will be a better way to exhibit their health impact than just merely stating it.
Round 2
Reviewer 1 Report
The manuscript organization is much improved. I have made very minor edits to the revised version as comments and using track changes.

Reviewer 3 Report
The author has addressed my questions and concerns, thus I think this manuscript can be accepted in present form.Author Response
Dear Reviwer 3,We again thank You again for the constructive and helpful comments.
With kind regards and best wishes,
Your sincerely,
Dr. Ana Miklavčič Višnjevec
Koper 6th August 2020